# OperatorEYEVP: Operator Dataset for Fatigue Detection Based on Eye Movements, Heart Rate Data, and Video Information

**DOI:** 10.3390/s23136197

**Published:** 2023-07-06

**Authors:** Svetlana Kovalenko, Anton Mamonov, Vladislav Kuznetsov, Alexandr Bulygin, Irina Shoshina, Ivan Brak, Alexey Kashevnik

**Affiliations:** 1Institute of Cognitive Neuroscience, HSE University, Moscow 101000, Russia; sv.d.kovalenko@gmail.com; 2Faculty of Physics and Mathematics and Natural Sciences, Peoples’ Friendship University of Russia, Moscow 117198, Russia; anton.mamonov.golohvastogo@mail.ru; 3Federal Research Center “Computer Science and Control” of Russian Academy of Sciences (FRC CSC RAS), Moscow 119333, Russia; ivladnet@ya.ru; 4St. Petersburg Federal Research Center of the Russian Academy of Sciences (SPC RAS), St. Petersburg 199178, Russia; alexandr_bulygin@mail.ru; 5Institute for Cognitive Research, Saint Petersburg State University, St. Petersburg 199034, Russia; shoshinaii@mail.ru; 6Faculty of Information Technologies, Novosibirsk State University, Novosibirsk 630090, Russia; i.v.brak@gmail.com; 7Institute of Mathematics and Information Technologies, Petrozavodsk State University, Petrozavodsk 185910, Russia

**Keywords:** fatigue, dataset, eye tracking, face and head video, gaze tracking, HRV (heart rate variability)

## Abstract

Detection of fatigue is extremely important in the development of different kinds of preventive systems (such as driver monitoring or operator monitoring for accident prevention). The presence of fatigue for this task should be determined with physiological and objective behavioral indicators. To develop an effective model of fatigue detection, it is important to record a dataset with people in a state of fatigue as well as in a normal state. We carried out data collection using an eye tracker, a video camera, a stage camera, and a heart rate monitor to record a different kind of signal to analyze them. In our proposed dataset, 10 participants took part in the experiment and recorded data 3 times a day for 8 days. They performed different types of activity (choice reaction time, reading, correction test Landolt rings, playing Tetris), imitating everyday tasks. Our dataset is useful for studying fatigue and finding indicators of its manifestation. We have analyzed datasets that have public access to find the best for this task. Each of them contains data of eye movements and other types of data. We evaluated each of them to determine their suitability for fatigue studies, but none of them fully fit the fatigue detection task. We evaluated the recorded dataset by calculating the correspondences between eye-tracking data and CRT (choice reaction time) that show the presence of fatigue.

## 1. Introduction

Fatigue has a huge impact on a wide range of everyday activities. We went through several statistical surveys dedicated to fatigue influence. According to a survey on reasons for accidents among truck drivers in India, fatigue and sleepiness caused 38% of all car accidents [1]. Another survey conducted in India states that a quarter of truck drivers reported that they were overspeeding during their assignments because of fatigue and tiredness [2]. They also mentioned that fatigue and lack of sleep is their biggest problem during working hours [3]. In Germany, in 2021, almost 30% of respondents pointed out that they had experienced tiredness and lack of concentration two to three times a week [4]. According to Department for Transport (UK), from November 2006 to July 2013, the number of drivers who continued to drive despite high levels of tiredness was never lower than 35% [5]. A high level of alertness and vigilance is indispensable for the error-free functioning of operators in the workplace. The main factors of attention loss are fatigue and drowsiness. Tasks performed by the operators are often routine, which also leads to a decrease in the level of attention [6]. High workload and monotonous tasks, excessive working hours, and night-time shifts lead to fatigue occurrence [7].

General reduction of working capacity caused by fatigue is proven to be accompanied by loss of attention, a decrease in reaction time, and an increase in the number of mistakes. As evidence, such mistakes lead to devastating consequences in the following working fields: air traffic control, nuclear power plant, and driving different vehicles (ships, aircraft, trains, trucks, public, and personal transport). In order to prevent emergencies, it is important to develop an early detection system for fatigue signs. In recent years, worker assistance systems are gaining popularity [8]. The authors analyze operator support systems based on eye-tracking data. They specifically outline that results are replicable and could be obtained in real-time. Whereas the authors note that the widespread introduction of such systems still involves a number of problems.

Unintentional human error in the workplace, which can include operator, maintenance, and management errors, is the most frequently identified cause of accidents, accounting for 30–90% of all serious accidents in various industries [9,10]. In aviation, there are rules according to which worker fatigue is controlled. In land transport, there are no parameters or methods for analyzing driver fatigue. On the part of the mental sphere, chronic fatigue, and especially overwork, appears dysfunctional of attention. Its stability and switching speed decrease, concentration is disturbed, and the amount of attention narrows. The functioning of operative memory worsens, thought processes slow down, and the function of forecasting, and foreseeing the situation suffers. There is a decrease in volitional efforts, and endurance and self-control are violated. Mental tension develops under the influence of both mental and physical factors. This leads to a change in the threshold of excitability of the nervous system and thereby interferes with the functional comfort of the operator. Fatigue detection using an eye tracker is a powerful non-invasive method. For early detection of fatigue, it is important to take into account the dynamics of changes. To evaluate this phenomenon, it is necessary to take into account several processes that occur simultaneously. Currently, there is no unified model for assessing the degree of fatigue, so data collection is an important task to create a comprehensive dataset that includes information from different sensors that monitor people in different states (normal and fatigued).

The main results of the research presented in the paper are as follows: Methodology development for presented dataset recording that is based on existing datasets analysis;Recorded dataset description and analysis that show the data consistency and makes possibilities for readers to understand how the dataset can be used in their research.

Presented a dataset combines several types of data signals that are synchronized in time. For each of our 10 participants, we have recorded 8 days of data 3 times a day. We have collected eye-tracking data, heart rate variability data, video recordings of head movements, a series of questionnaires performed every day and every session, and also, several scales performed before dataset recording. In each session, participants competed in the following tasks: choice reaction time (CRT) at the beginning and the end of the session, reading scientific-style text, correction test “Landolt rings”, and playing the game “Tetris”. Each session lasted for approximately 1 h. It is possible to mark up eye-tracking data using other types of signals and rely on it for further analysis. Currently, there are no publicly available datasets that would provide such data signals together. Thus, it is possible to use both its individual components (for example, heart rate parameters) and their combination. The recorded data can be used to search for relationships between eye movements and the functional state of fatigue. A variety of tasks performed by the participant have a complex load on the organism during the day. Long-term recording (8 days) provides analysis of data in dynamics and eliminates fluctuations in data within one day. Our dataset also includes a fatigue score for each participant, allowing comparison of subjective (scales) and objective (eye and head movements, heart rate variability) measurements.

## 2. Related Work

We have reviewed datasets related to the paper topic that are publicly available. The main conditions for including the dataset in the studied list were as follows:Data recording using an eye tracker (or camera);Open access to raw or processed data;Description of methodology and type of data processing, as well as structure of data storage.

The data listed below meet these requirements and may also contain additional data signals, such as EEG, EOG, and video recording of the scene camera.

Gaze in wild. This dataset was recorded in contemplation of analyzing eye and head movements in the performance of everyday tasks [11]. Participants performed the following actions: catch a ball, search for an object, brew tea, and perform indoor navigation. This dataset contains a set of images and videos that were recorded with Pupil Labs eye tracker, accelerometer, gyroscope IMU MPU-6050, and ZED stereo camera. The following parameters were obtained: eyes and head rotation velocity, infrared images of eyes, and images of scenes (RGB + D), gaze fixations (GF), tremor, drift, microsaccades, fixation of the rotational vestibular-ocular reflex (rVOR), fixation of the translational vestibular-ocular reflex (tVOR), gaze tracking, gaze shift. A manual data markup was performed (140 min in total) for further neural network training.

Dr(eye)ve. The aim of this dataset is an estimation of attention shifts during driving [12]. Moreover, the evaluation of the outside vehicle environment, as well as the dynamics of attention between the in-vehicle and out-vehicle environment, is taken into account to form the context. The participants were driving a car under the following conditions: time of the day (3 times of the day), type of weather conditions, and type of terrain. SMI Eye Tracking Glasses 2 Wireless (SMI ETG 2w) glasses were used to record the data. The dataset consists of fixations and their temporal integration, as well as objects on which they were detected (roads, signs, trees, etc.).

EEGEyeNet. The objective of the dataset [13] is to improve the understanding of brain and eye functioning. The dataset consists of two synchronized types of signals—EEG and eye tracking data (128-channel EEG Geodesic Hydrocele system, ET EyeLink 1000 Plus). Participants were asked to perform the following three types of tasks: pro- and antisaccade paradigm (fixation on the target stimulus in the right and opposite directions), large grid paradigm (sequential fixation on a set of points), the paradigm of visual symbol search (search for the target stimulus line by line). The data were annotated, and the following three types of events were distinguished: fixations, saccades, and blinks. In total, 356 people took part in the experiment; the total amount of data are 47 h.

EYE-EEG. This dataset [14] consists of four subsets collected for various purposes. The authors studied the task of classifying emotional facial expressions using EEG, EOG, and eye tracking (SMI IView X high-speed eye tracker). The second and third datasets were also recorded under conditions of simultaneous recording of EEG and eye tracking data (SMI IView X tracker and Eyelink 1000 tracker, respectively). In the second group, the task was to search for the target image, and in the third—to read the list of words. The fourth group is under development; it is aimed at checking the synchronization of the TX-300 Tobii Pro equipment. All these datasets are processed using the EYE-EEG tool, which allows working with an electroencephalogram and eye tracker simultaneously.

LEDOV. The purpose of creating the dataset [15] was to study human attention drawn by a video clip. In total, 32 participants took part in the experiment and watched 538 videos. Eye movement data were recorded using Tobii TX300. More than five million fixations were detected from obtained data. The developers came to the following conclusions: there is a high correlation between the probability that an image contains an object and that this object attracts human attention. Human attention is more likely to be drawn by moving objects or moving parts of objects. Moreover, there is a temporal correlation between human attention and a smooth transition of saliency across video frames.

LPW (labeled pupils in the wild). The dataset [16] is based on high-quality, high-speed videos of the eye area. It was established for the development and evaluation of pupil detection algorithms. In total, 22 participants who represented five different nationalities, were following moving objects with their eyes in different locations. To record eye movement data, the Pupil Pro head-mounted eye tracker was used, and the PointGrey Chameleon3 USB3.0 camera was selected as the scene camera. Next, the authors evaluated the performance of the five most popular pupil detection algorithms. Parameters such as accuracy and reliability, type of lighting, glasses, and makeup, different resolutions were also taken into consideration.

EyeC3D. The dataset [17] was created to study the mechanisms of the visual system. It has been addressed within the framework of visual attention theory that controls viewing 3D content. This is one of the few publicly available datasets that provide such a stimulus. The dataset consists of eight stereoscopic video sequences. It also contains information about fixation points and fixation density maps. The movements of both eyes were recorded separately from each other using two computers and an eye tracker Smart Eye Pro 5.8.

DOVES (a database of visual eye movements). The dataset [18] was recorded to create an open database of high-precision eye movements. Twenty-nine participants viewed about a hundred images. All images with artificial structures, animals, faces, and others have been excluded. To eliminate fixations at the same point, participants were asked to perform a simple memory task that facilitated viewing the entire image. The recording was made with Dual-Purkinje Eye Tracker (Gen 5), and the database contains about 30,000 fixation points.

Jetris. The dataset [19] was developed to study the detection of cognitive fatigue in teachers using eye tracking. In the conditions of competition or collaboration, sixteen participants played “Tetris” for 5 min. Both oculometric data and the results of games were recorded. Based on four parameters (average pupil diameter, the standard deviation of the pupil diameter, saccade rate, and number of fixations over 500 ms), the cognitive fatigue index was calculated.

VQA-MHUG. The dataset [20] was collected during Visual Question Response (VQA) to analyze the similarities between human and neural attention strategies. Attention maps based on 4000 question-image pairs were generated for 49 participants. To register oculomotor activity, the EyeLink 1000 Plus was used.

MASSVISS. This dataset [21] was collected to study visualization (websites, journals, blogs, etc.). The authors examined image elements in order to study their influence on memory, perception, and understanding. For a detailed study of visualization, authors created a taxonomy that classifies images based on their content. Thirty-three participants in the experiment viewed 393 visualizations. Fixation coordinates and durations, as well as the scanpath, were recorded. In total, about 600 fixation points were received for one visualization (one visualization was shown for about 10 s). The data were recorded using SR Research EyeLink1000 desktop eye-tracker.

CrowdFix. This dataset was collected in order to study visual attention in terms of the following two types of stimuli: static and dynamic. The authors aimed to evaluate several existing saliency models. This dataset contains eighty-nine videos of crowds with three levels of density. Twenty-six participants viewed those videos, and their eye movements were recorded with the EyeTribe eye tracker. It also contains fixations, fixation maps, and saliency maps. They also calculated the number of fixations and average duration for each type of crowd density [22].

Who is Alyx? This dataset was recorded for biometric data research. It was collected during playing a virtual reality game “Half-Life: Alyx”. It contains the following several types of data signals: body and eye movements, controller interactions and screen recording, ECG, EDA, PPG, IBI, body temperature, HRV, and acceleration. Seventy-one participants played this game for two days for 45 min. Eye movements were recorded with Unity and SRanipal. It includes raw eye tracking data, such as gaze direction, the position of pupils, etc. [23].

The-way-of-the-future. This dataset not only includes eye movement data but also motion data. It was collected to obtain a non-laboratory dataset. It includes several everyday tasks, such as walking, climbing stairs, and others. For 1.5–2 h NN, participants were recording their actions using the Pupil Labs eye tracker. Raw data are available that can later be preprocessed with Pupil Labs software [24].

Table 1 presents a comparison of datasets. An analysis showed that, nowadays, there is no set of data that can be used for studying operator fatigue. Some of them, for example, Gaze in wild, have all the necessary information about gaze parameters and head movements, but participants performed several physical activities, which makes this dataset irrelevant in terms of the study of operator fatigue. The same problem occurs with datasets Who is Alyx? Additionally, the-way-of-the-future. Virtual reality affects the physical and mental condition of participants. Datasets [13,14,16,18,20,21] also contain tasks that are not close to everyday assignments at a working place. Dataset Jetris meets most of the requirements, but unfortunately, the duration of performed tasks is not long enough to cause fatigue.

## 3. Experimental Methodology

We have developed a methodology that obtains a set of data to study the phenomenon of operator fatigue as a dynamic state. Simultaneous recording of several data streams at different times of the day helps to obtain a more detailed picture of the development of this condition. Moreover, it provides related data (reaction time and heart rate variability) about what processes are happening in the human body. Different types of tasks are required since when performing the same type of assignments, it is impossible to track the dynamics of this phenomenon. Recording data for eight days allows us to track whether these processes have built-in dynamics, periodic processes, or cycles.

The experimental setup records the following data:Eye movement;Head movement;Image of the scene;Heart rate (PPI);Choice reaction time (twice);Questionnaires and scales.

To objectify the data obtained as evidence of a state of fatigue, a choice reaction time (CRT) was recorded. CRT is one of the types of Go/No-Go tasks. It characterizes neurodynamic processes in the central nervous system. Theoretically, by the end of the working day, there is an increase in reaction time of CRT by an average of 30% [25]. To reduce the number of errors, it is proposed to perform CRT measurements repeatedly. It allows us to calculate stability of the reaction based on the standard deviation, which increases with fatigue.

Figure 1 shows the scheme of the experimental setup that was used for the registration of physiological and behavioral indicators of operator fatigue. Table 2 includes parameters of apparatus used in experimental setup.

There are several parameters that should be taken into account when studying fatigue. In particular, those related to the amount and quality of sleep that determine the possibility of recovery after a hard day.

Amount of sleep in the last 4 h;Cumulative sleep deficit;Number of waking hours since the last main sleep period;Time of day on the biological clock;Number of changed time zones.

Cumulative sleep deficit should not exceed 10 h; otherwise, cognitive decline will occur [26]. If a person is awake for 17 or more hours, level of fatigue becomes dangerous. If a person works during his/her usual sleep hours, this can also lead to a critical level of fatigue. The same effect is given by changing time zones for more than three hours [26]. We used Pittsburgh Sleep Quality Index Questionnaire (PSQI) to track the influence of sleep factor on functional state. It contains questions about the quality and duration of sleep and possible sleep disorders [27].

To exclude participants with clearly expressed signs of chronic fatigue, we used Fatigue Assessment Scale (FAS) [28].

To exclude participants with depressive symptoms, the Beck Depression Scale (BDI-II) was used. Depressive symptoms affect cognitive functioning and attention (and attention indicators related to global/local information analysis [29]. This questionnaire allows measuring severity of depression [30]. Depression also affects activity of the dorsal/ventral system, ambient/focal strategy of eye movements, and development of fatigue. Participants who showed symptoms of depression based on this scale were not invited to take part in dataset recording.

The proposed methodology takes into account the key factors of individual differences that may affect the parameters of the eye movement strategy. In particular, it is a field-dependent/independent cognitive style [31]. It is interconnected with the mechanisms of global and local information analysis and dominance of the right or left hemisphere [32]. Fatigue disturbs the balance between the methods of information analysis. Individuals with a field-dependent cognitive style analyze information from the general to the particular. On the other hand, people with a field-independent style analyze from particular to general [33]. The method of “Gottschaldt Figures” [34] is classically used to assess field dependence.

Tests for the dominant eye and hand [32] also make it possible to take into consideration individual characteristics of information perception. They are related to field-dependence/independence and global/local mechanisms of information analysis, dorsal/ventral system, ambient/focal strategy eye movements, and dynamic/static vision.

For the brain, the information coming from the leading eye prevails over the information received from the non-leading eye when reconstructing an environmental image. It depends on the dominant eye, which hemisphere mainly receives information. With the left leading eye, we can say that most of the information enters right hemisphere. According to neurophysiological data [35], in this hemisphere, dorsal system dominates, which is associated with ambient strategy of eye movements and dynamic vision. Additionally, respectively, with the mechanism of global information analysis. There are two types of information analysis mechanisms—global and local. Depending on the type of structures being processed, the strategy of eye movements is determined—ambient or focal.

In order to conduct an experiment and analyze the state of operator fatigue, it was proposed to perform following psychological tests and scales. They take into account key individual features of information analysis and functional state that affect fatigue indicators (participants took all the listed psychological tests once before the start of the experiment) as follows:Gottschaldt’s Hidden Figure Test (GHFT), field dependence-independence;Ocular dominance test [36];Handedness questionnaire determines severity of motor asymmetry [32];Beck Depression Scale (BDI-II) [30];Pittsburgh Questionnaire for Determining the Sleep Quality Index [27];Fatigue Assessment Scale (FAS) [28].

For daily assessment of current state of fatigue (as well as indicators of the quantity and quality of sleep) VAS-Fatigue (VAS-F) and a list of questions about the quality of sleep over the past night were used [37]. The VAS-F scale consists of 18 questions related to the subjective perception of fatigue and is filled in before each entry in the experimental session (3 times a day).

Two types of tasks were used for developing the scheme of experimental session. They are based on different response styles—passive and active. Specifically, the following:Controlling engagement and level of attention (reading a scientific-style text);Requiring a coordinated motor response (CRT, correction test “Landolt rings”, Tetris).

Registration of CRT includes the following stages: presentation of a target stimulus (red circle with a diameter of 2 cm) and a distractor (green circle with the same radius). The task was to press button when target stimulus appeared (70 trials). The following parameters were recorded: average reaction time, standard deviation, and number of errors. Due to the fact that operator could have different levels of fatigue at the beginning and at the end of the recording session, it was decided to conduct the CRT test twice (at the beginning and end of the experimental session).

Participants were asked to read scientific-style text in order to imitate everyday working tasks. It acts as a control condition and load static task. We used articles and books found in an indexed database. The task was performed for 15 min with laptop or computer screen.

Correction test “Landolt rings” is a test used for measuring visual acuity (stimuli—rows with rings resembling letter ‘C’). The width of the ring gap is one-fifth of its total diameter and can be at any place (total 8). Participant has to indicate all the rings where the desired gap is located in 5 min in table (30 × 30) [38]. The following parameters were recorded: the time spent (if participant completed the task in less than 5 min), number of target stimuli, number of detected and missed stimuli, number of errors, and number of elements and lines viewed. Based on these parameters, indicators of attention productivity, work accuracy, stability of attention concentration were calculated. As well as coefficient of mental productivity, efficiency, concentration of attention, amount of visual information, and speed of processing.

Tetris is used as a control condition and load dynamic active task. It is an arcade game for studying hand-eye coordination. Participants were asked to achieve the best result under 15 min. We recorded number of games, scores, levels, and lines [39].

Scheme of experimental session consists of the following:Sleep quality questionnaire (once a day);VAS-F questionnaire (three times a day);Choice reaction time (Go-no-go paradigm);Scientific-style text;Correction test “Landolt ring”;Game “Tetris”;Choice reaction time (Go/no-go paradigm).

Registration of eye movement data, video, and PPI recording was carried out simultaneously. After each of the tasks, recording was stopped. Figure 2 shows the sequence of actions for one session. The total duration of such recording was around 1 h.

Figure 3 shows the sequence of actions during one day of data recording, which consists of three sessions (at 9:00, 13:00, and 18:00). Participant fills out a daily diary of the recordings to document session’s features (errors in equipment, distractors).

Figure 4 shows the chronology of recording and complete dataset for one participant. It consists of records of at least seven days.

## 4. Dataset Overview

Figure 5 shows the structure of the collected dataset of all participants in the “Experiments” folder. This folder contains other folders (experiment content) that include the participant’s id. The name of the participant’s folders contains the date of recording. The “Experiments” folder contains the “Eyes” folder, “PPI” folder, and “Videos” folder, which includes eye tracker data, PPI data, and webcam videos, respectively. There are also files “Metadata” and “Summary”. File “Metadata” includes several sheets reporting results of the CRT task (reaction time, standard deviation, number of errors), results of the correction test (19 parameters), results of the Tetris game (overall score, score, lines, levels, completed game, duration) and parameters of heart rate variability (58 parameters) and results of questionnaires scoring (BDI_II, FAS, PSQI, dominant eye, and hand) and also general information about the participant (age, gender). The file “Summary” includes all this information and quantitative gaze characteristics. Each session has its own CRT task results, correction test “Landolt rings” results, Tetris game results, and HRV indexes.

Figure 6 shows the structure of the “Eyes” folder, which contains folders with eye-tracking data information for each session (five records of morning, afternoon, and evening sessions, respectively). Each folder contains a .csv file with gaze coordinates and a .mp4 file with scene footage. The gaze coordinates file contains the gaze fixation timestamp, number of frames, horizontal gaze coordinate, and vertical gaze coordinate.

Figure 7 shows a scene footage example when the participant is completing a reading task. The text is presented on the laptop screen, and the green circle represents the participant’s gaze.

Table 3 shows an example of a .csv file with gaze coordinates. It contains parameters such as gaze_timestamp, world_index, confidence, norm_pos_x, and norm_pos_y. Parameter gaze_timestamp is a timestamp of the source image frame. Parameter world_index is the number of world video frames. Parameter confidence is computed confidence between 0 (not confident) and −1 (confident). Parameter norm_pos_x is x position in the world image frame in normalized coordinates. Parameter norm_pos_y is y position in the world image frame in normalized coordinates [40].

Figure 8 shows the structure of the “Videos” folder containing five records of morning, afternoon, and evening sessions, respectively. The folders include .mp4 files from the webcam.

Figure 9 shows the structure of the “PPI” folder. The “PPI” folder contains five records of morning, afternoon, and evening sessions, respectively. Time of day folders contains .txt files with PPI values representing cardiac pulse-to-pulse intervals.

As mentioned in the methodology section characteristics, including CRT task results, correction test “Landolt rings” results, Tetris game results, HRV indexes, and psychological tests are presented in the .xlsx file “Metadata” that is accessible for each session of experiments. These characteristics describe the psychophysiological state of a person when recording a session.

Table 4 shows the “CRT” sheet of the .xlsx file “Metadata”. It contains CRT task results: average reaction time, standard deviation, and errors. These data represent the following two stages: at the beginning (stage 1) and at the end (stage 2) of the session in the morning, afternoon, and evening recordings according to the presented experiment methodology.

Table 5 shows the “Corr_test” sheet of the .xlsx file “Metadata”. It contains correction test “Landolt rings” results, 19 parameters in sum. The day column is the time of the day when the session has been recorded, T is the time spent, n is the total number of characters to be crossed out, M is the total number of characters crossed out, S is the correctly selected letters, P is the letter characters missed, O is the wrongly selected characters, N is the total number of characters viewed up to the last selected character, C is the total number of lines viewed, A is the indicator of the speed of attention (attention performance), T1 is the work accuracy indicator (first option), T2 is the work accuracy indicator (second option), T3 is the work accuracy indicator (third option), E is the mental productivity index, Au is the mental performance, K is the concentration (percentage of correctly highlighted characters from all that needed to be highlighted), Qualitative is the qualitative characteristic of concentration, Ku is the indicator of stability of attention concentration, V is the capacity of visual information, Q is the processing rate.

Table 6 shows the “Tetris” sheet of the .xlsx file “Metadata”. It contains the Tetris games results. Overall score is the sum of scores for all games. Scores is the number of points scored in one game. Lines is the number of constructed lines. Level is the number of levels. Completed is the game ended before time ran out. Duration is the exact total playing time.

Table 7 shows the “HR” sheet of the .xlsx file “Metadata”. It contains HRV indexes calculated based on PPI interval values for the participant using the software Kubios [41]. There are 58 characteristics in total, which fall into the following three categories: time-domain, frequency-domain, and nonlinear HRV analysis methods.

Table 8 shows the “Scales” sheet of the .xlsx file “Metadata”. It contains results of BDI-II, PSQI, FAS, Gottschaldt Figures, dominant eye, and hand questionnaires for Participant 8. Moreover, it includes general information about participants—age and gender.

Table 9 represents the result example of the VAS-F scale for every day of recording for Participant 8. It contains the sum of all answers for each time of the day for two subscales (fatigue and energy). Table 10 includes an example of information about the average quality of sleep and mean time to fall asleep for Participant 8 (results of sleep quality scale).

Every participant folder contains the .xlsx file “Summary”. File “Summary” is an Excel table that contains CRT task results, correction test “Landolt rings” results, Tetris game results, heart rate variability indexes, and quantitative characteristics of gaze.

First goes general data about the record—the file name, a method for determining numerical characteristics, performed activity, and time of the day. The next columns indicate measured values of CRT—reaction time, mean deviation, and the number of errors for recordings made at the beginning and at the end of the session. The next go characteristics obtained from the performing correction test “Landolt rings”, a total of 19 characteristics.

The following columns represent the results of the Tetris game: total score; the score for one game, lines and level achieved per game for each of the three sessions; whether the last game was completed, and for how long the whole session lasted. Empty cells in columns indicate the absence of a second and/or third attempt.

Next goes heart rate variability data. The number of columns differs for each participant due to the different number of epochs extracted from PPI files. Cells containing “NaN” are non-computable numbers.

Cells containing “none” indicate the absence of this characteristic for this record. Next are calculated numerical characteristics of the identified oculomotor strategies, 60 in total. All characteristics were recorded in accordance with the proposed methodology. Table 11 shows characteristics that are highly correlated with the average reaction time and standard deviation of CRT.

Reaction time 1 and Deviation 1 are the average reaction time and standard deviation at the beginning of the session. Reaction time 2 and Deviation 2 are the average reaction time and standard deviation at the end of the session. Speed is gaze speed in degrees of visual angle. Sac < 6° is the number of saccades with an amplitude less than 6 degrees of visual angle. Max PTM is the maximum ratio of path-to-movement of the gaze.

## 5. Dataset Analysis

In this section, we describe the analysis of all types of signals we have collected in this dataset. It includes analyses of eye movements, heart rate variability, metadata (questionnaires scores, CRT reaction time, parameters of correction test, and results of Tetris game), and videos.

### 5.1. Eye Movements Characteristics

We have calculated a number of characteristics based on information of eye movements. In order to suppress noise and minimize outliers, characteristic values were filtered using the percentile function of 10. This value of the filtering threshold was used since, at a lower value, the noise clearly affected the data. Based on it, speed, acceleration, quantity, time, percent, frequency, ratio, and length characteristics were calculated.

The correlation of quantitative characteristics with average reaction time and standard deviation of the CRT task was calculated. Table 12 shows characteristic types and the number of characteristics used. In total, 60 characteristics of 8 types were calculated.

Calculation of the average path length to displacement ratio of the gaze (average_path_to_displacement) is shown in Figure 10. On the timeline, the symbol “f” is fixation, and the symbol “s” is saccade. Distance traveled by the gaze is the path length. The displacement of the gaze is a vector that connects the initial and final positions of the gaze. The path length to displacement ratio is calculated in 4-s intervals. The path (path_1_) is divided by the displacement (displacement_1_) in the first 4-s interval, and the path-to-displacement ratio (path_to_displacement_1_) is obtained. Then the path-to-displacement ratio (path_to_displacement_2_) is calculated for the second 4-s interval. The minimum (min_path_to_displacement) and maximum (max_path_to_displacement) path-to-displacement ratios are found by iterating over all path-to-displacement ratios. All path-to-displacement ratios over all 4-s intervals are summed (all_path_to_displacement) and divided by their number (path_to_displacement). Thus, the average path-to-displacement ratio is obtained.

The calculation of the duration of fixations shorter/longer than 180 ms (t_less_180_ms/t_more_180_ms), the percentage of fixations shorter/longer than 180 ms (fix_percentage_less_180/fix_percentage_more_180), number of fixations shorter/longer than 180 ms per minute (num_fix_less_180_min/num_fix_more_180_min) are presented below. If the distance from the fixation to the center of the fixation area d_c_ is less than or equal to the fixation area radius (d_c_ ≤ r_A_), then the gaze is inside the fixation area (Figure 11).

The time (Δt) that the gaze was inside one fixation area is counted. If Δt is less than 180 ms, then the Δt is added to t_less_180_ms, and the number of fixations shorter than 180 ms (num_fix_less_180) increases by one, otherwise the Δt is added to t_more_180_ms and the number of fixations longer than 180 ms (num_fix_more_180) increases by one. Then num_fix_less_180/num_fix_more_180 is divided by the total duration of record in minutes (all_time_minutes) to receive num_fix_less_180_min/num_fix_more_180_min, and num_fix_less_180/num_fix_more_180 are divided by the number of all fixations (all_fix) to receive fix_percentage_less_180/fix_percentage_more_180. The minimum/maximum frequency of appearance of a new fixation area per second (min_fix_freq/max_fix_freq) and average frequency of appearance of a new fixation area per second are calculated as follows. The number of fixations in each second interval is calculated, and min_fix_freq/max_fix_freq are found by iterating over them. The average frequency of appearance of a new fixation area per second is calculated by dividing the number of all fixations by the number of all second intervals. Minimum/maximum saccade length (min_sac_l/max_sac_l) and average saccade length are calculated as follows. All saccade lengths are calculated, and min_sac_l/max_sac_l are found by iterating over them. The average saccade length is calculated by dividing the sum of all saccade lengths by their number.

The calculation of the minimum/maximum eye movement speed in a second interval (min_speed/max_speed) and the module of minimum/maximum acceleration in the second interval (min_acc/max_acc) are presented below. The gaze path (distance_1_) and the travel time (time_1_) are calculated for the first second interval (Figure 12). On the timeline, the symbol “f” is fixation, and the symbol “s” is saccade. Next, distance_1_ is divided by time_1,_ and the speed for the first time interval (speed_1_) is obtained. Similarly, the speed for the second interval is calculated (speed_2_), etc. The minimum/maximum eye movement speeds in the second interval are calculated by iterating over all speeds. The speed at the beginning of the gaze equal to zero is subtracted from the speed_1,_ and the result is divided by time_1_. Then the module from the result is taken, and thus the acceleration module for the first second interval is obtained (acc_1_). The speed_1_ is subtracted from the speed_2,_ and the result is divided by time_2_. After that, the module from the result is taken, and thus the acceleration module for the second second interval is obtained (acc_2_). Similarly, the acceleration module for the third second interval is calculated (acc_3_), etc. The min_acc/max_acc are calculated by iterating over all acceleration modules. The average gaze speed in a second interval (average_speed) is calculated by dividing the path for all time intervals (all_distance) by the number of all second intervals (sec_intervals). The module of average acceleration in the second interval (average_acc) is calculated by dividing the sum of all acceleration modules (acc_sum) by sec_intervals.

From 60 quantitative characteristics, 23 were selected that have a stable correlation greater than or equal to 20%. This value was chosen since at correlation values less than 20%, they rapidly decrease. Table 13 shows the characteristic’s name, correlation of characteristics with average reaction time and standard deviation of CRT task, and diameter of gaze fixation area. The diameter of the gaze fixation area is a value that has been taken to calculate gaze fixation (in degrees of visual angles).

### 5.2. Scales and Questionnaires

In this section, we present an analysis of subjective scales that participants have filled in during dataset recording. Table 14 shows experiment participants with information about all questionnaires and scales performed beforehand. It also includes information about the gender and age of participants. Among all participants, there is one subject who showed symptoms of depression (Participant 3). We decided to include this participant in our dataset for reference. Participants 1 and 2 were excluded from further analysis because the first participant had symptoms of depression, and the second one had contact lenses. All other participants had no symptoms of depression, chronic fatigue, or sleep problems. Most of the participants showed field dependence, which means that most of them analyzed information from general to particular.

Every day before starting the morning recording, participants answered questions about sleep quality during the previous night. Below we present Table 15, which includes mean sleep quality through all of the dataset recordings, as well as mean time to fall asleep. As we can see, participants reported that the quality of their sleep during dataset recording was fairly good or fairly bad. The mean time to fall asleep varies between 11.8 (Participant 8) minutes and 72.2 (Participant 7) min.

Next, we present the result of processing the VAS-F scale. This scale contains two subscales that assess the degree of fatigue and energy. Figure 13 shows the dynamics of changes in the degree of fatigue for all of the participants. As can be observed, the subjective feeling of fatigue is lower in the afternoon than in the morning and evening. Participant 4 (red) shows the highest scores and growth during the day. Participant 3 (blue) also has relatively high scores, while Participants 6 (pink) and 10 (khaki) have the lowest. In Figure 14, we present the energy scores of this scale. Participant 4 (red) has the sharpest decline during the day. The highest scores for the energy subscale appear to be in the afternoon.

For better visualization, we present mean scores for both of the subscales in Figure 15. Peak energy values appear to be in the middle of the day. The lowest fatigue values consequently are in the afternoon.

### 5.3. CRT

Below are presented graphs of CRT results. They include information about the dynamics of changes in the average reaction time (solid lines) and standard deviation (dotted lines) during the day and the measurement period. Graphs represent data of 9 participants for 8 days of recording. There are three sessions and two CRTs—at the beginning (on the left side) and at the end (on the right side) of each session.

Figure 16 represents morning sessions. According to it, the average reaction time at this time of the day is between 302 and 594 ms. The standard deviation varies between 28 and 135 ms. Participant 3 (blue) has a noticeably longer reaction time. This participant showed symptoms of depression according to the BDI-II scale. Participants 4 and 8 (red and grey lines, respectively) show the best performance among other participants. Most of the days, both of them complete this task worse at the end of the session than at the beginning (50 ms on average). Generally speaking, CRT reaction time at the end of the session takes a smaller range of values.

Figure 17 represents reaction time and standard deviation in the afternoon. The average reaction time is between 292 ms and 617 ms. The standard deviation is between 25 ms and 119 ms. The same participant shows the best values in both measurements (Participants 4 and 8). Participant 3 shows the worst reaction time at this time of the day.

Figure 18 represents evening sessions. The average reaction time is between 300 ms and 556 ms. The standard deviation is between 20 ms and 136 ms. The same participant shows the best results for both of the CRTs (Participants 4 and 8). Participant 3 has the highest values among other participants.

We can notice that all values of CRTs stay in one range, which means that our dataset is consistent.

Figure 19 represents the difference between reaction time at the end and at the beginning of each session for all of the participants for 8 days. It accepts positive and negative values. Positive values mean that reaction time at the beginning of the session was better than at the end. Negative values mean the opposite. As we can see, the widest ranges between reaction times are in the morning and in the evening (the lowest point is −103 ms and the highest is 101 ms), whereas in the afternoon, it is the smallest and takes values around zero except for several days. As it might be seen, participants have different patterns throughout the whole dataset recording their results vary within days. Participant 3 shows the biggest number of negative values among other participants. Participants 4 and 10, for most of the days, completed CRT worse at the end of the session.

Figure 20 represents values of CRT reaction time at the beginning of each session for all of the participants for 8 days. Based on this graph, we can say that our participants fall into the following two categories: those who perform better in the afternoon and those who perform worse at this time of the day. For most of the participants, we observed better reaction time in the evening compared to the morning. Participant 3 (blue) has the biggest reaction time (617 ms), while Participant 4 (red)—has the smallest (292 ms). For some of the participants, we did not observe any dependencies, and for some, they were multidirectional.

Below is presented Figure 21 with CRT at the end of the session. As we can see, results drastically change in the middle of the day.

Then visualizations of the mean reaction time for CRT at the beginning (CRT_1) and at the end (CRT_2) of the session (Figure 22) were calculated for all participants as mean values. As we can see, participants perform CRT_1 better in the middle and at the end of the day.

It is important to note that the mean reaction time at the end of the session does not change that drastically. Figure 23 represents this tendency.

CRT results are sensitive to the time of the day. We also observe individual differences between participants.

### 5.4. PPI

We have calculated 58 indexes using Kubious software based on PPI data see (Appendix A). These indexes reflect the activity of the sympathetic and parasympathetic nervous systems. The approaches used to analyze PPI fall into the following three categories: time-domain, frequency-domain, and nonlinear HRV analysis methods. The first category is calculated based on beat-to-beat PPI interval values. For the second category power spectrum density is calculated for the PPI interval series. The last category includes non-linear mechanisms of heart rate regulation. These methods are mainly based on standards of measurement provided by the Task Force. Calculation of indicators for each time of day and activity was carried out in the .txt file. In this file, the second column of data contained the value of the time in ms between PP intervals. The data were loaded into the Kubios software. Next, a filter was selected to correct artifacts in the “strong” configuration. After that, depending on the duration of the recording, the entire recording was divided into fragments of 5 min each. If the duration of the fragment was less than 3 min, then it was not taken into account. Moreover, initially, the entire record was evaluated for artifacts, and if there were too many of them, the record was excluded from the analysis. After that, all indicators were calculated in a file with the csv extension. Further, the general base of the study was compiled from the obtained data.

### 5.5. Video

Our dataset contains a video of the operator’s face and torso. Based on this video, using modern computer vision technologies, we determined the following operator characteristics: respiratory rate [42], heart rate [43], blood pressure [44], oxygen saturation [45] as well eyes/mouth openness and closeness [46] and head/body movements using our developed earlier human analyzing tool. The tool marks every second of the video by vital signs determined using the methods presented in our previous papers mentioned above. In Figure 24 result of the markup is presented.

### 5.6. Correction Test “Landolt Rings”

In this section, we present several parameters of the correction test “Landolt rings”. To evaluate the accuracy of performing a test, the total number of crossed-out characters must be divided by the number of characters that should have been crossed out. This is how the work accuracy indicator is calculated. It is presented below in Figure 25 for all of the participants for 8 days. As we can see, the best performance for all participants appears to be in the afternoon, whereas we observe a decline in the morning and the evening. Moreover, we observed that range of values becomes smaller over time, which is explained by the learning effect. Participant 11 completed this task much worse than other participants (magenta line).

The mental productivity index is calculated as follows: the total number of characters viewed up to the last selected character is multiplied by the work accuracy indicator. This index is presented below in Figure 26. As demonstrated on the graph, most of the time mental productivity index is the highest in the afternoon. Consequently, Participant 11 (magenta) shows lower values of this parameter.

The next parameter is concentration. It represents the percentage of correctly highlighted characters. Here, in Figure 27, we see the same trend as for the mental productivity index and work accuracy indicator. The best concentration mostly occurs in the afternoon. Values vary between 30 (Participant 11) and 100 (Participants 8 and 10). It means that Participants 8 and 10 made this task completely correct.

For all of the participants, all three parameters are graphically pretty similar to each other. However, we observe several differences. For example, some participants (e.g., Participants 3, 4, 5, 11) have an overall better work accuracy indicator than the mental productivity index. Concentration and work accuracy indicators, on the other hand, are completely the same, minor differences are explained by different scales.

Finally, we present the indicator of stability of attention concentration in Figure 28. It is based on the total number of lines viewed, missed, and wrongly selected characters. Accordingly, Participants 8 and 10 had the highest values of this indicator, several times during recording they achieved the best possible result—zero mistakes with total completeness of the task. Those peaks take values of 900 due to the specificity of the calculation. Participant 11 has the lowest results among other participants. Moreover, we can point out that the stability of attention concentration increases as days go.

General conclusion according to correction test results. It is useful in terms of fatigue development, indicating that the best performance occurs in the afternoon.

### 5.7. Tetris

Below, in Figure 29, are the presented results of the Tetris game. To compare the scores of all participants, we divided their overall scores at each session on the duration of the game. We call this measurement the Tetris index. The first thing that can be noticed is that all of the participants show an increased level of skill in playing Tetris over time. It is explained by learning, which occurs because of the repetitiveness of the task. Increased scores in the evening also could be explained by learning. For further analysis, we exclude scores from the first day to eliminate this effect. Again, Participant 11 shows the worst results in this task. This can be explained by the fact that this person is significantly older than other participants. At the same time, this participant performed poorly only in Tetris and Landolt; his CRT reaction time was average. Whereas Participant 9 shows the best results among other participants, especially on day 6. This could happen due to the fact that he is field-independent. Participant 4, who showed the best results in CRT, is also one of the best Tetris players. Participant 3, with symptoms of depression, performs worse on the CRT task but completes the correction test and plays Tetris at an average level. Participants mostly have two kinds of patterns during dataset recording. The first one is characterized by top performance in the afternoon, and the other one is the opposite.

When comparing the results of Tetris and CRT throughout the recording of the entire dataset, we can identify the following trend: skill to play the game improves, as does the reaction time.

## 6. Discussion

Fatigue state is a diverse, non-linear phenomenon that includes 2–3 components. To detect and evaluate it, we need an ecologically valid model. People get tired differently at different times of the day, which requires data recording throughout the entire day. We brought the dataset closer to reality and looked from different points of view, such as taking into consideration the amount of sleep, subjective fatigue assessments, and results of performed tasks.

Perhaps a sufficient degree of fatigue is not achieved in one session because we are trying to achieve an ecologically valid model of human behavior. This means that it is necessary to use diverse tasks within the framework of the developed methodology. The methodology we have developed has an important aspect that allowed us to avoid the influence of monotony. We used Tetris as a type of task that differs in its dynamics from other similar tasks presented. Tetris is a non-standardized type of task; however, it provides quantitative results of a game. The peculiarity of this game lies in its competitive effect, which can subsequently affect the performance of CRT. That is, after the end of the game of Tetris, we can observe the following effects: an improvement in reaction time and a reduction in the standard deviation compared to the CRT at the beginning of the session. The developed methodology and collected data should help to confirm or refute the following hypotheses.

Subjective feelings of participants correlate with objective indicators;Person should be tired during different times of the day (morning, afternoon, evening);State of fatigue should be correlated with CRT (choice reaction time) and heart rate variability measurements;Eye movement characteristics should be indicators of fatigue.

We analyze the presented dataset to study data consistency. We can conclude that CRT values lie within the same corridor both for one participant and for the entire sample. The obtained results of the analysis of the VAS-F scale correlate with each other; that is, the subscales of energy and fatigue have the opposite form. Several of Landolt indicators (work accuracy indicator, mental productivity index, concentration) look almost identical, which indicates that the results of this test are stable. Based on our findings, future research according to the collected dataset can confirm or refute the mentioned hypotheses.

We also identify the following limitations to the recorded dataset.

Lack of environmental data, which usually includes data obtained from the study of operator behavior;We did not control the ambient lighting conditions and brightness settings of the equipment used (e.g., face illumination is different from recording to recording);Learning effect of the operator is inevitable since, in methodology, we propose a long duration of data recording (that can be simply seen during the dataset analysis);Testing effect could appear for the VAS-F scale that was filled in three times a day; however, on the one hand, it may affect the resulting score; on the other hand, the visual analogue scale is a great tool for subjective assessment of feelings;Data synchronization has been implemented by timestamps, and since we used different devices for data recording, we can guarantee the synchronization of different signals by 1–2 s;We do not have older participants (more than 60 y.o) in the dataset, but it seems, for the considered tasks, we cannot see applications where retired people can work on key point operator positions.

## 7. Conclusions

In the scope of the paper, we reviewed datasets that have public access. Each of them was described in terms of the following points: the purpose of data collection, types of collected data, apparatus, type of performed task, the number of participants, the principle of dataset structure, and extracted characteristics.

We recorded the dataset with the following modalities: eye movements, video of the scene, video from web camera, heart rate variability, reaction time and standard deviation of choice reaction time task, results from Tetris game, results from correction task “Landolt rings” task and results of multiple questionnaires (BDI-II, PSQI, FAS, field dependence, VAS-F, and sleep quality for each session day). We have extracted the following data: (speed of gaze movement, module of acceleration, speed in the fixation area, saccade speed, and gaze velocity).

We also obtained a detailed profile of each participant using multiple questionnaires. In addition, we tracked their subjective feelings of fatigue throughout the experiment.

This dataset will form the basis for creating our own fatigue identification model. The distinctive features of this large-scale dataset are the variety of tasks performed, the total number of records, and the simultaneous recording of several types of signals. All this makes it unique in comparison with other datasets aimed at studying fatigue and which are in the public access. Based on this dataset, it is possible to study the dynamics of changes in this functional state from different points of view. In addition to objective ways of measuring it, one can also study their correlation with the subjective assessment of the participants themselves.

Operator fatigue detection is the future for fatigue routine control of employees. At the moment, there are a lot of startups that detect driver drowsiness in vehicle cabins. But in contrast, the fatigue detection approach presented in the paper provides the possibility of early detection of the fatigue state (before the driver closes his/her eyes).

In the future, this dataset can be implemented in many areas. We are going to prove or refute the formulated hypotheses as well as we are going to develop a software for high-precision detection of fatigue in the operator’s workplace for routine control of fatigue of operators (vehicle drivers, machine operators, etc.).

## Figures and Tables

**Figure 1 sensors-23-06197-f001:**
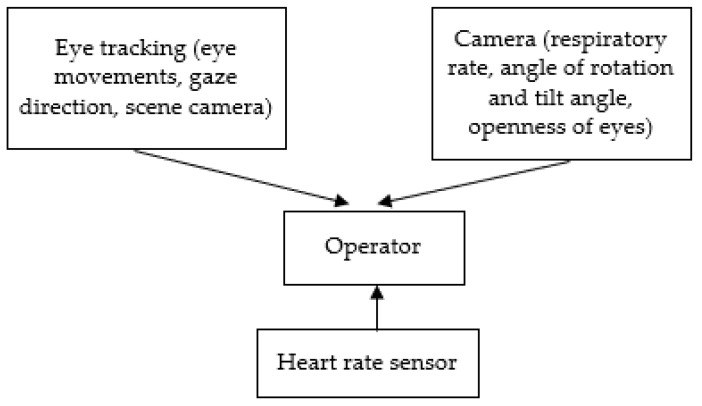
Scheme of experimental setup.

**Figure 2 sensors-23-06197-f002:**
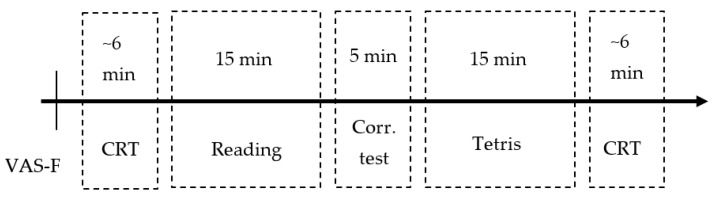
Timeline of one session.

**Figure 3 sensors-23-06197-f003:**
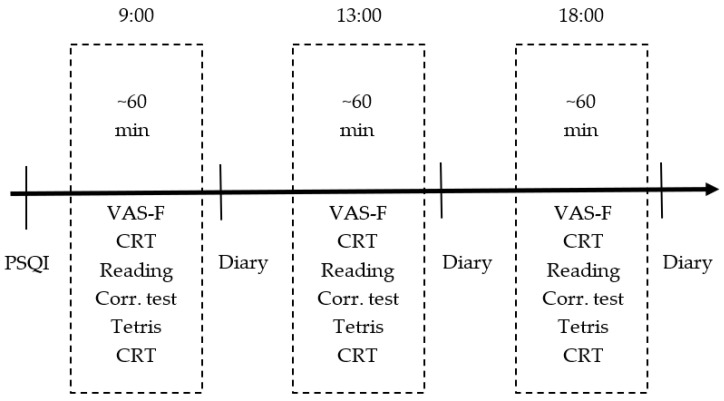
Timeline of one day of recording.

**Figure 4 sensors-23-06197-f004:**
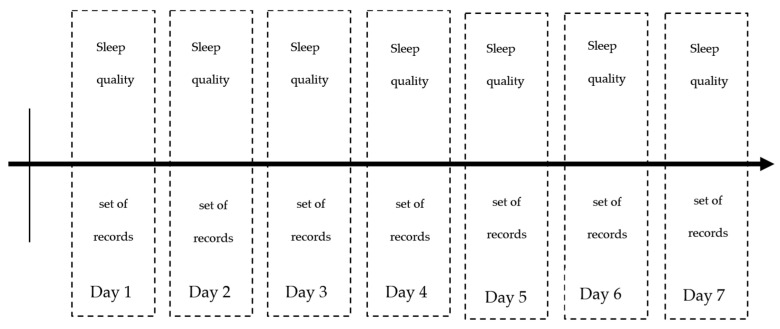
Timeline of dataset recording.

**Figure 5 sensors-23-06197-f005:**
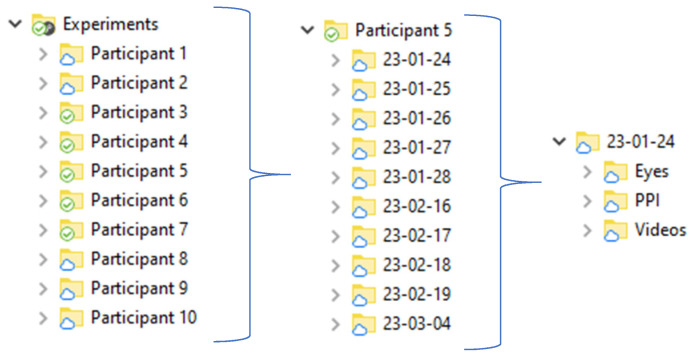
Structure of “Experiments” folder.

**Figure 6 sensors-23-06197-f006:**
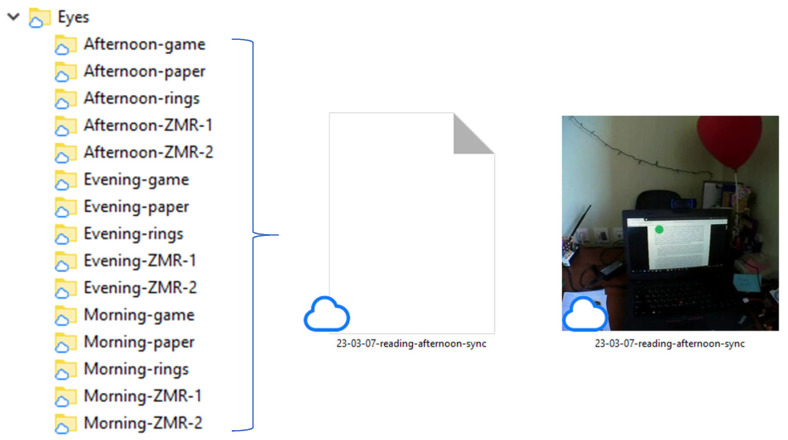
Structure of “Eyes” folder.

**Figure 7 sensors-23-06197-f007:**
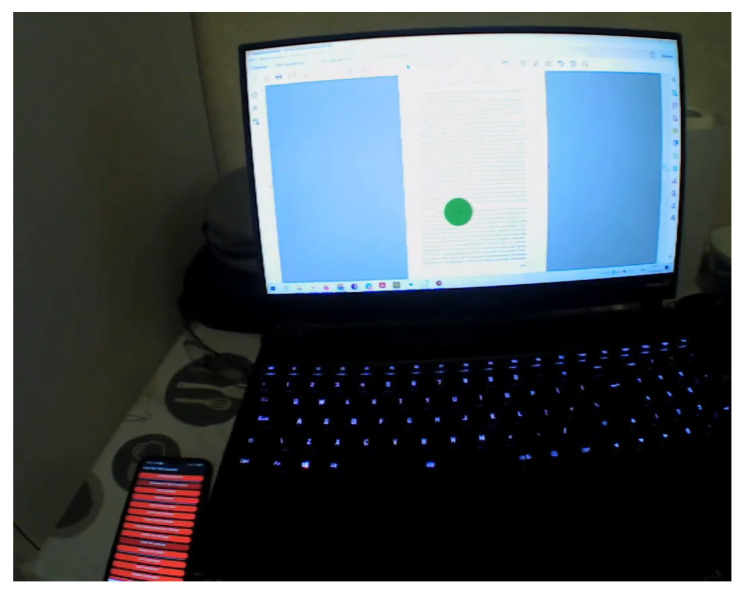
Scene footage example.

**Figure 8 sensors-23-06197-f008:**
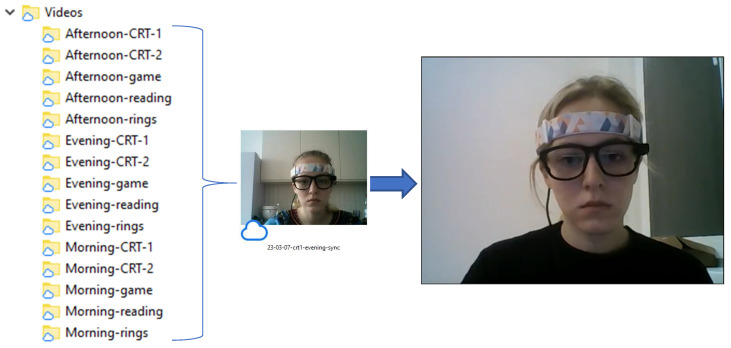
Structure of “Videos” folder.

**Figure 9 sensors-23-06197-f009:**
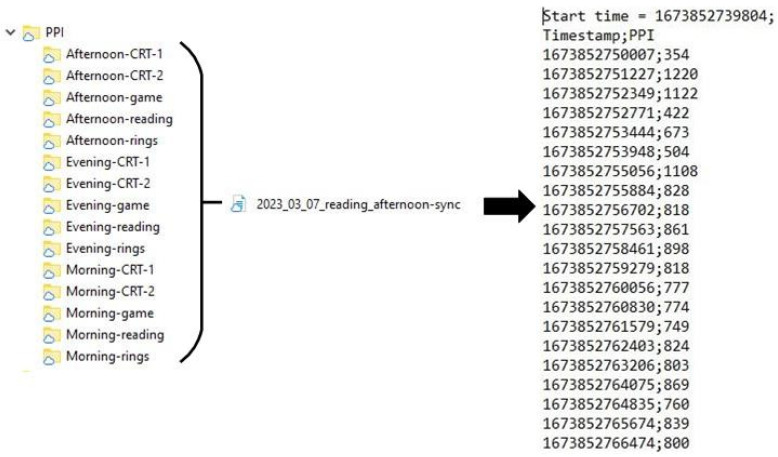
Structure of “PPI” folder.

**Figure 10 sensors-23-06197-f010:**
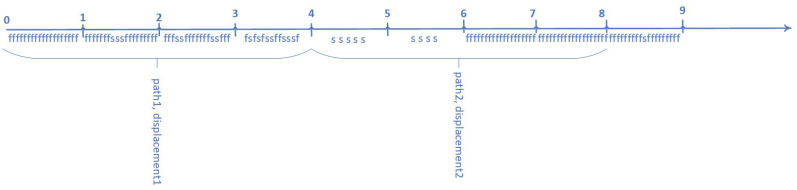
Calculating the path to displacement ratio (f—fixation, s—saccade; 0–9—seconds).

**Figure 11 sensors-23-06197-f011:**
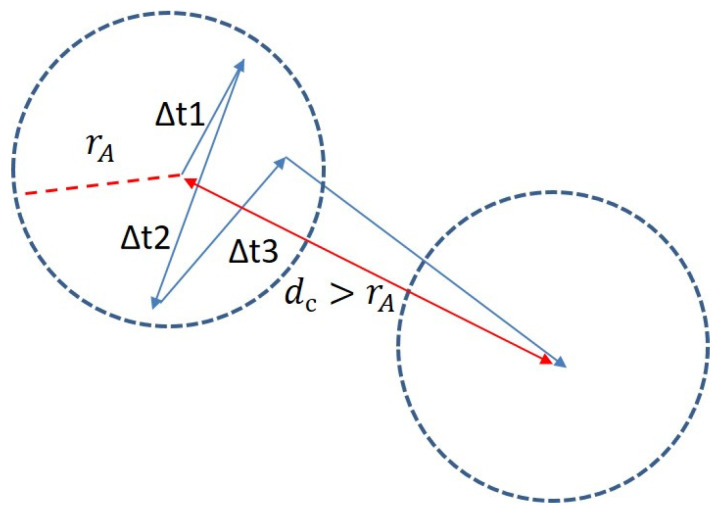
Fixation area and saccade.

**Figure 12 sensors-23-06197-f012:**
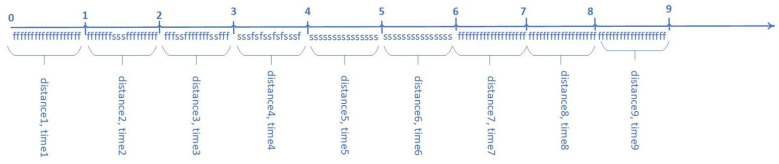
Speed and acceleration module in second interval (f—fixation, s—saccade, 0–9—seconds).

**Figure 13 sensors-23-06197-f013:**
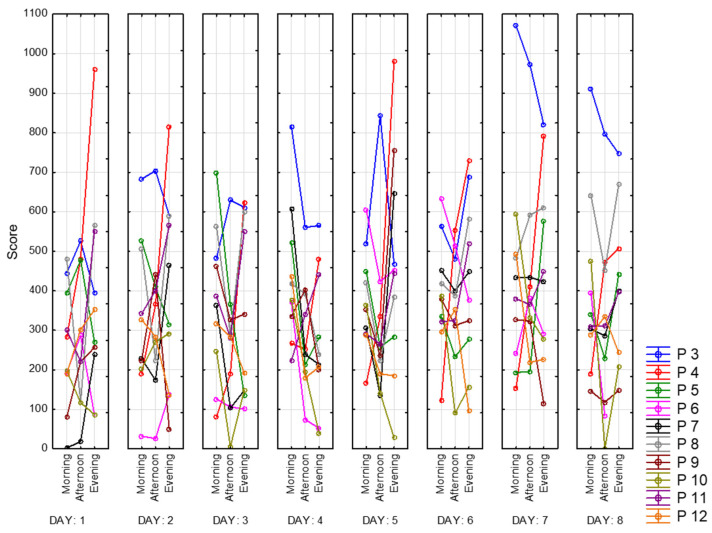
Fatigue subscale scores.

**Figure 14 sensors-23-06197-f014:**
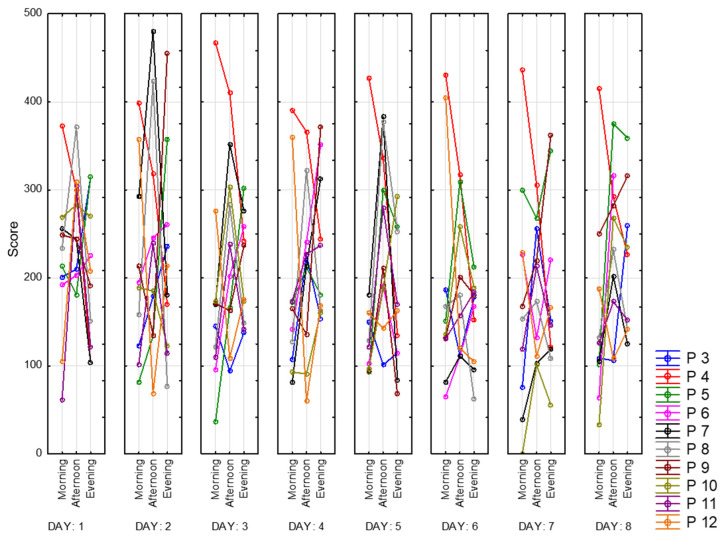
Energy subscale scores.

**Figure 15 sensors-23-06197-f015:**
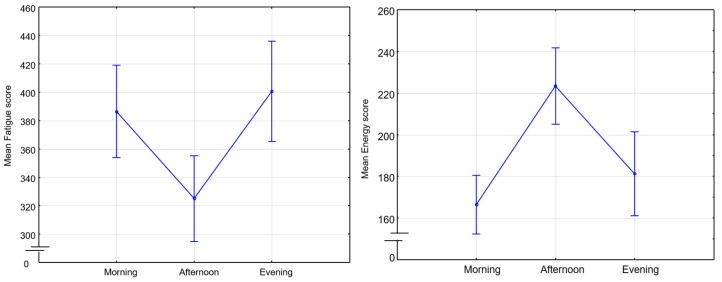
Mean scores of VAS-F subscales. Fatigue and energy.

**Figure 16 sensors-23-06197-f016:**
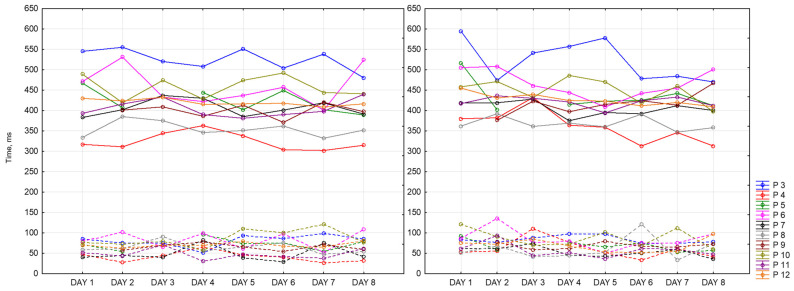
CRT at the beginning and at the end of morning session.

**Figure 17 sensors-23-06197-f017:**
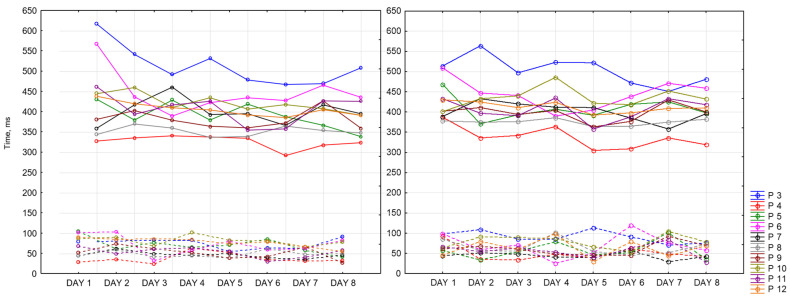
CRT at the beginning and at the end of afternoon session.

**Figure 18 sensors-23-06197-f018:**
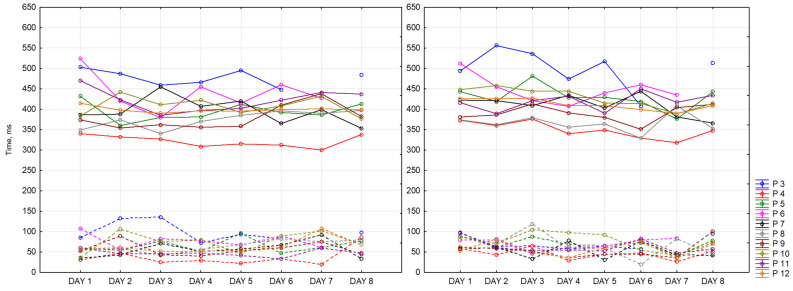
CRT at the beginning and at the end of the session.

**Figure 19 sensors-23-06197-f019:**
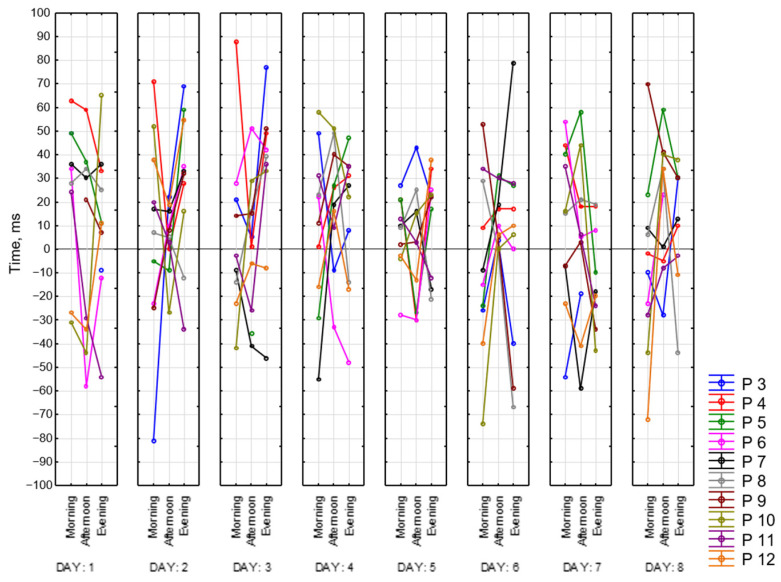
Difference between CRT at the beginning and at the end of each session.

**Figure 20 sensors-23-06197-f020:**
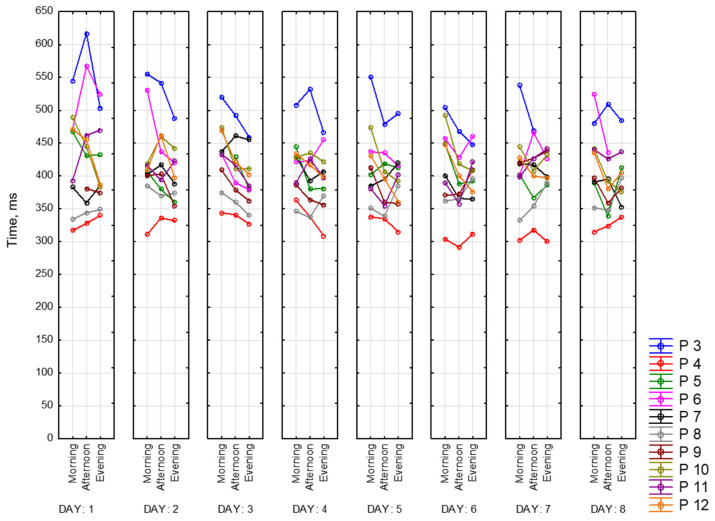
CRT at the beginning of each session.

**Figure 21 sensors-23-06197-f021:**
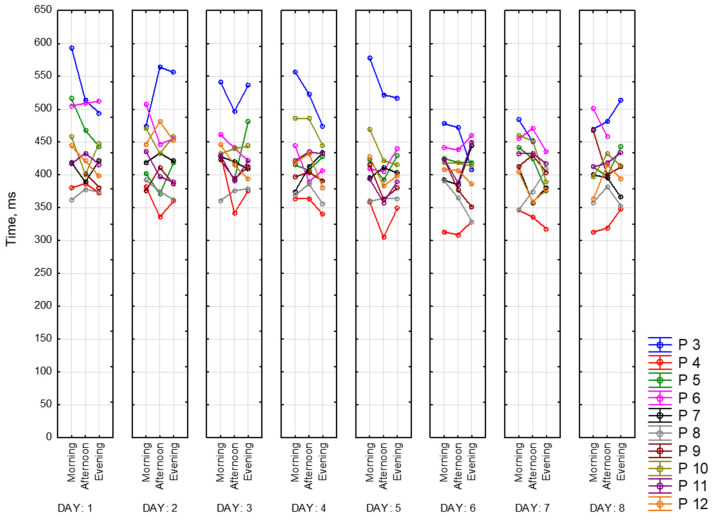
CRT at the end of each session.

**Figure 22 sensors-23-06197-f022:**
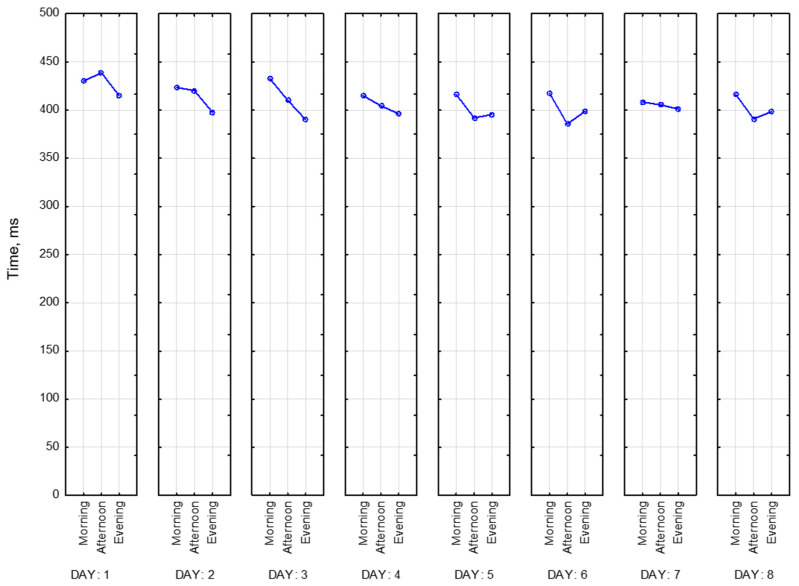
Mean reaction time at the beginning of the session.

**Figure 23 sensors-23-06197-f023:**
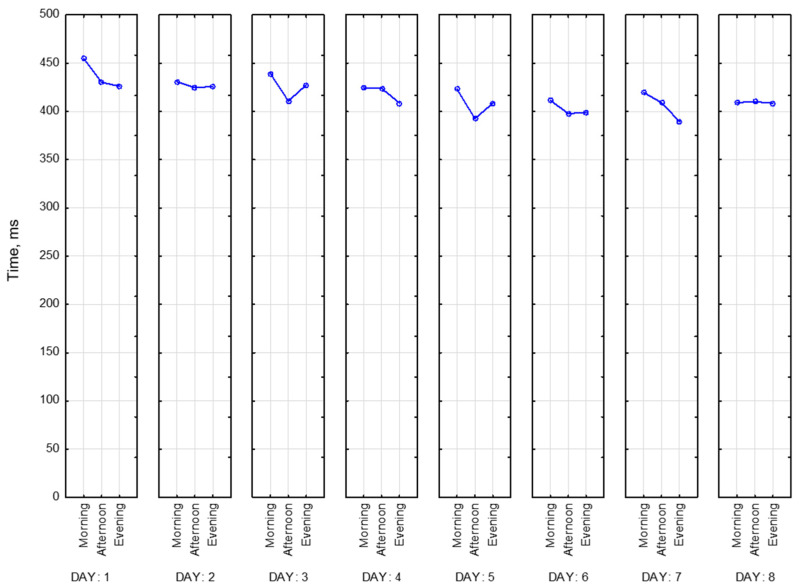
Mean reaction time at the end of the session.

**Figure 24 sensors-23-06197-f024:**
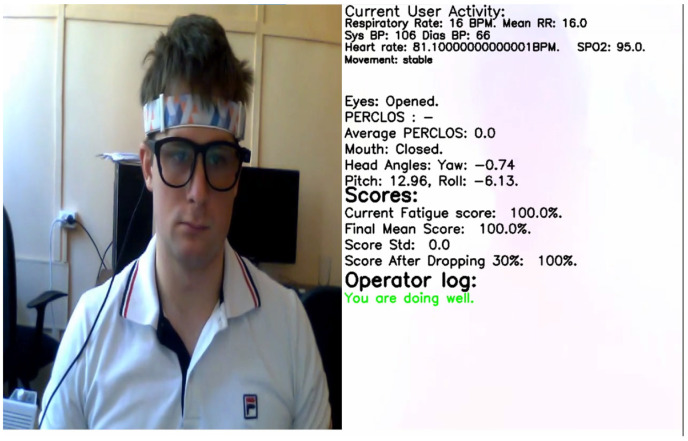
Example of a video marked by a human analyzing tool.

**Figure 25 sensors-23-06197-f025:**
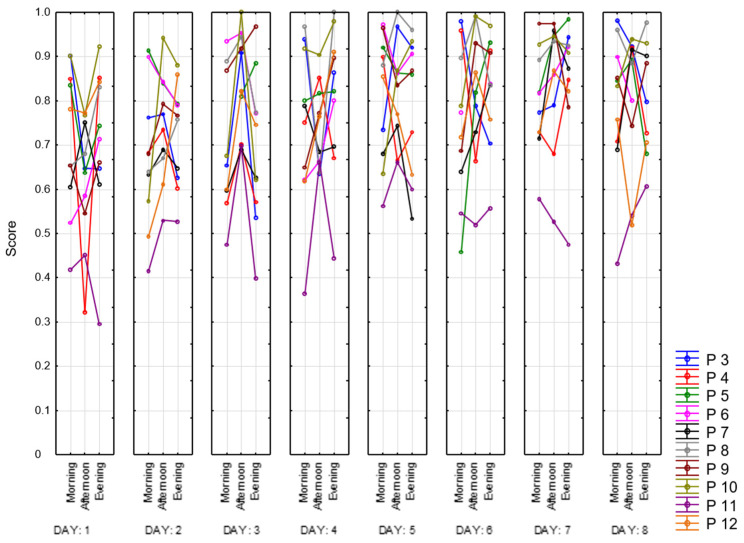
Work accuracy indicator.

**Figure 26 sensors-23-06197-f026:**
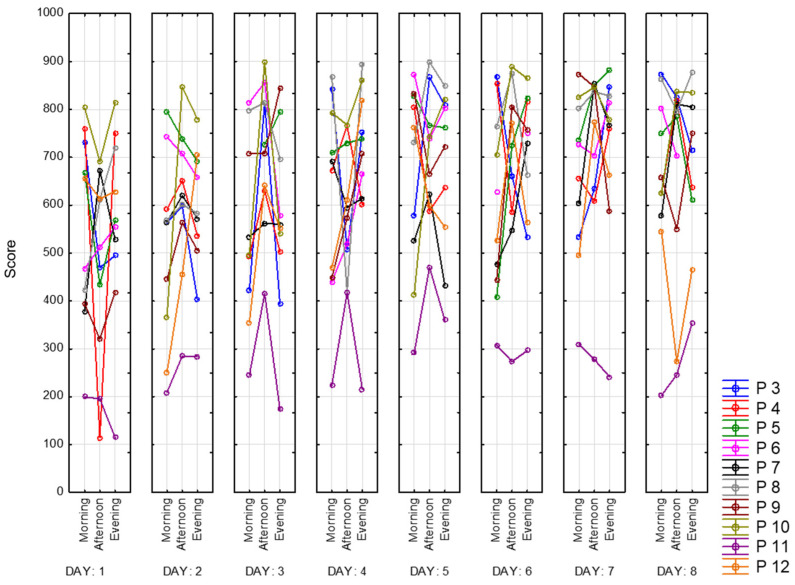
Mental productivity index.

**Figure 27 sensors-23-06197-f027:**
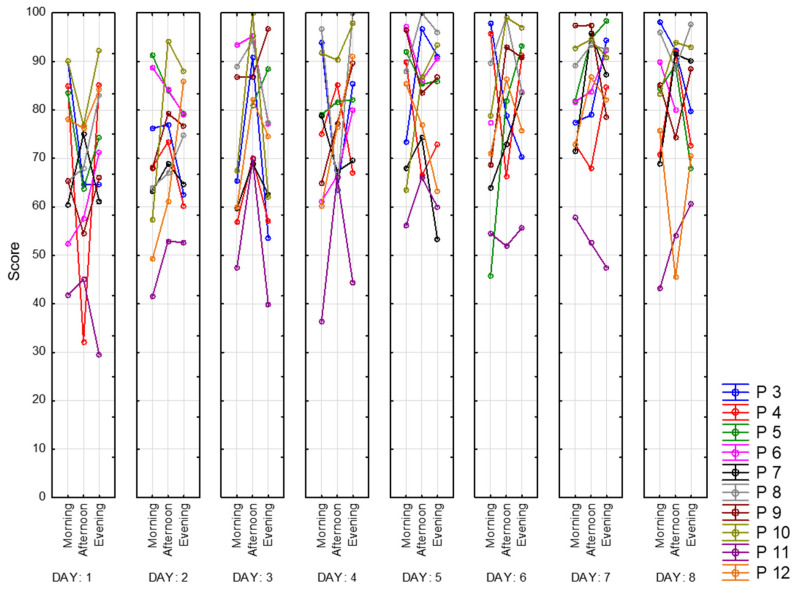
Concentration.

**Figure 28 sensors-23-06197-f028:**
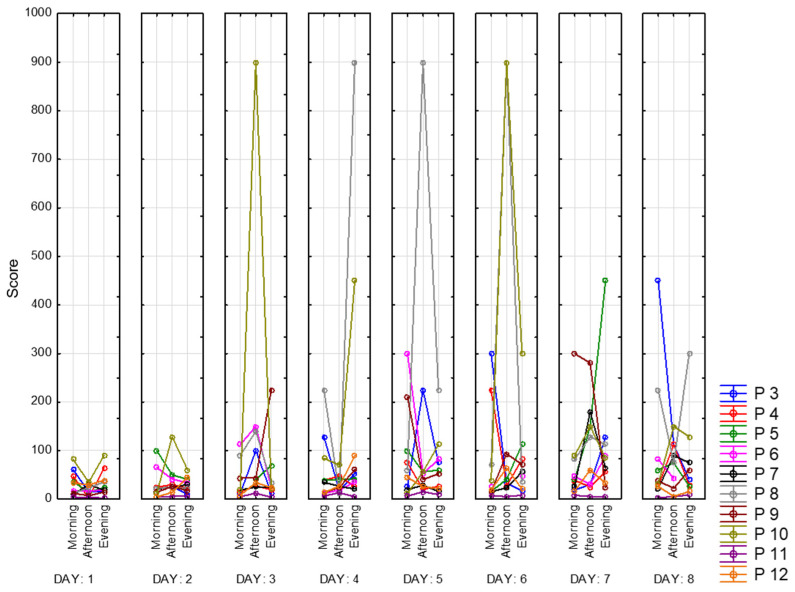
Indicator of stability of attention concentration.

**Figure 29 sensors-23-06197-f029:**
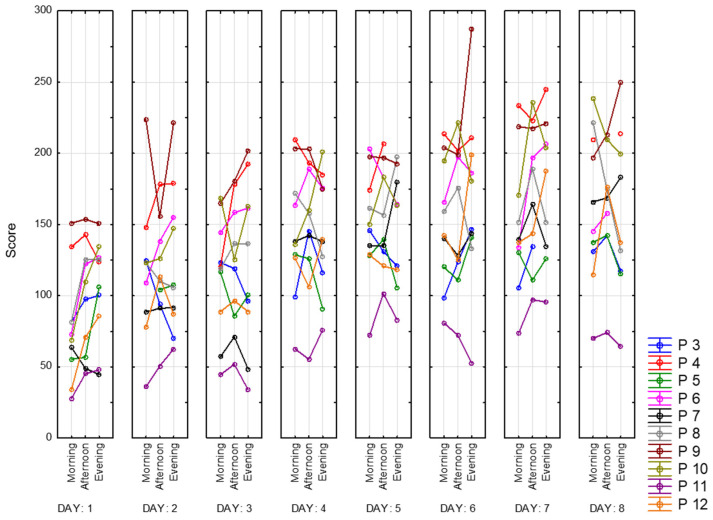
Tetris index.

**Table 1 sensors-23-06197-t001:** Description of publicly available dataset.

Dataset	Eye Tracking Hardware	Performing Tasks	Recording Length, (Hours, Frames)	Characteristic
Gaze in wild [11]	Pupil labs	Everyday tasks	23 h	19 thousand fixations, 18 thousand saccades, 13 thousand smooth pursuit, 3.5 thousand blinks
Dr(eye)ve [12]	SMI Eye Tracking Glasses 2 Wireles	Driving	>6 h	5 hundred thousand of frames, fixations, and its temporal integration
EEGEyeNet [13]	EyeLink 1000 Plus	Pro- and anti-cascade paradigm, large grid paradigm, paradigm of visual symbol search	47 h	Fixations, saccades, blinks
EYE-EEG [14]	SMI iView X hi-speed Eyelink 1000 Tobii TX300	Classifying emotional facial expressions, search for the target image, read the list of words.	Unknown	Gaze coordinates, pupil size, eye coordinates, pupil diameter
LEDOV [15]	Tobii TX300	Watching video	More then 1.5 h	5 million fixations
LPW [16]	Pupil Pro head-mounted eye tracker	Following moving object with eyes	22 min	66 videos of eye area
EYEC3D [17]	Smart Eye Pro 5.8	Watching stereoscopic videos	Unknown	Unknown
DOVES [18]	Dual-Purkinje Eye Tracker (Gen 5)	Viewing images	Unknown	30 thousand of fixation points
Jetris [19]	-	Playing Tetris	About hour and a half	Fixation points, saccades
VQA-MHUG [20]	EyeLink 1000 Plus	Viewing images, answering questions	Unknown	12 thousand gaze samples
MASSVIS [21]	EyeLink100	Viewing images	Unknown	Fixation coordinates and duration, scanpath
CrowdFix [22]	EyeTribe	Viewing videos	Unknown	Around 23 thousand fixations, fixation, and saliency maps
Who is Alyx? [23]	SRanipal	Playing game in VR	110 h	Gaze direction, pupil position
The-way-of-the-future [24]	Pupil Labs	Everyday tasks	24 h	Fixations, gaze coordinates

**Table 2 sensors-23-06197-t002:** Elements of experimental setup.

Apparatus	Parameters
Eye tracker	Pupil invisible glasses—200 Hz. Scene camera—30 Hz. Gyroscope, accelerometer
Heart rate sensor	Polar verity sense. Heart rate per minute, photoplethysmography values, PP interval, accelerometer, gyroscope, magnetometer
Camera	Min 640 × 480, 30 fps
Monitor	Diagonal 15.6″, IPS 1920 × 1080

**Table 3 sensors-23-06197-t003:** Example of gaze coordinates file.

Gaze_Timestamp	World_Index	Confidence	Norm_Pos_x	Norm_Pos_y
2.363712504	0	1.0	0.520421589122099	0.4268529821325231
2.365695504	0	1.0	0.5206671883078182	0.4266634340639468
2.373860504	0	1.0	0.521640889784869	0.42482791476779513
2.377724504	0	1.0	0.5215625762939453	0.42535739474826384
2.381705504	0	1.0	0.5207383211921243	0.4260887993706597
2.385669504	0	1.0	0.5178640589994543	0.4263024224175347
2.393702504	0	1.0	0.5147253485286937	0.42662466543692135
2.397653504	0	1.0	0.5133877922506893	0.4263908103660301
2.401730504	0	1.0	0.5124125761144301	0.42617916531032984
2.405666504	0	1.0	0.5114908218383789	0.4261219731083623
2.363712504	0	1.0	0.520421589122099	0.4268529821325231

**Table 4 sensors-23-06197-t004:** Example of CRT task results that is included to Metadata file.

Time of Day	Stage	Average Reaction Time	Standard Deviation	Errors
Morning	Stage 1	449	76	1
	Stage 2	425	76	3
Afternoon	Stage 1	388	86	2
	Stage 2	419	51	1
Evening	Stage 1	392	47	3
	Stage 2	419	73	3

**Table 5 sensors-23-06197-t005:** Example of correction test “Landolt rings” results.

Day	t	n	M	S	P	O	N	C	A	T1	T2	T3	E	Au	K	Qualitative	Ku	V	Q
Morning	301	105	48	48	57	0	892	30	2.96	0.45	0.45	0.45	407.7	−0.2	45.7	Average	15.7	529	1.2
Afternoon	302	121	99	99	22	0	886	30	2.93	0.81	0.81	0.81	724.9	1.8	81.8	Very good	40.9	525	1.5
Evening	301	119	111	111	8	0	884	30	2.93	0.93	0.93	0.93	824.5	2.5	93.2	Very good	112.5	524	1.6

**Table 6 sensors-23-06197-t006:** Example of the Tetris game results.

Time of Day	Overall Score	Game 1			Game 2			Completed	Duration
		Scores	Lines	Level	Scores	Lines	Level		
Morning	122,000	115,000	90	9	7000	5	0	No	15:54
Afternoon	136,000	136,000	106	10				Yes	16:15
Evening	105,000	28,000	24	2	77,000	64	6	No	16:34

**Table 7 sensors-23-06197-t007:** Example of several HRV indexes.

Date	Time of Day	Process	S1_PNS Index	S2_PNS Index	S3_PNS Index	S1_SNS Index	S2_SNS Index
16 February 2023	Morning	CRT-1	−0.82176			0.867574	
16 February 2023	Morning	reading	−0.81644	−1.22041	−1.30499	1.207893	1.548793
16 February 2023	Morning	rings	−0.98766			1.393083	
16 February 2023	Morning	game	−0.72016	−1.18462	−1.17657	0.961316	1.464698
16 February 2023	Morning	CRT-2	−0.80627			0.973839	
16 February 2023	Afternoon	CRT-1	−1.95545			2.777188	
16 February 2023	Afternoon	reading	−1.83054	−1.74177	−1.98811	2.425533	2.151486
16 February 2023	Afternoon	rings	−1.71012			2.670973	
16 February 2023	Afternoon	game	−2.1294	−2.01461	−2.04313	3.577055	3.039821
16 February 2023	Afternoon	CRT-2	−1.55164			1.83855	
16 February 2023	Evening	CRT-1	−1.38388			1.494551	
16 February 2023	Evening	reading	−1.25131	−1.46615	−1.14954	1.220413	2.021247
16 February 2023	Evening	rings	0.089249			0.234475	
16 February 2023	Evening	game	−1.67382	−1.77946	−1.5725	2.435569	2.543351
16 February 2023	Evening	CRT-2	−1.63793			1.862338	

**Table 8 sensors-23-06197-t008:** Example of questionnaires scores for Participant 8.

Age	Gender	BDI-II	PSQI	FAS	Gottschaldt Figures	Dominant Eye	Dominant Hand
22	f	8	6	24	field-dependent	right	right

**Table 9 sensors-23-06197-t009:** Example of VAS-F results for every day of dataset for Participant 8.

Day	Morning Fatigue	Morning Energy	Afternoon Fatigue	Afternoon Energy	Evening Fatigue	Evening Energy
5 March 2023	481	233	117	371	566	151
6 March 2023	505	158	224	423	588	77
7 March 2023	563	121	290	283	599	149
8 March 2023	418	128	392	322	240	163
9 March 2023	420	129	223	377	384	253
10 March 2023	419	168	387	180	582	63
11 March 2023	483	153	592	173	611	108
12 March 2023	642	132	453	232	669	141

**Table 10 sensors-23-06197-t010:** Sleep quality example for Participant 8.

Mean Sleep Quality	Mean Time to Fall Asleep
Fairly bad	11.8 (min 5, max 25)

**Table 11 sensors-23-06197-t011:** Example of quantitative characteristics of gaze.

Date	Time of Day	Activity	Average Reaction Time before, ms	Standard Deviation before, ms	Average Reaction Time after, ms	Standard Deviation after, ms	Number of Saccades with Amplitude of Less Than 6°/min	Maximum Curvature of the Trajectory	Average Speed of Gaze
25 January 2023	Morning	CRT 1	407	55	402	59	16.2	71.8	20.9
25 January 2023		Reading					11.1	39.6	25
25 January 2023		Rings					7.1	11.7	14.4
25 January 2023		Game					15.2	29.9	27.6
25 January 2023		CRT 2					22.2	70.7	25.9
25 January 2023	Afternoon	CRT 1	380	60	371	34	12.1	23.2	19.9
25 January 2023		Reading					11.6	41.7	24.7
25 January 2023		Rings					8.2	25.6	15.2
25 January 2023		Game					15.2	56.8	31.3
25 January 2023		CRT 2					15.1	29.7	23.4
25 January 2023	Evening	CRT 1	360	55	419	61	8.1	26.1	17.5
25 January 2023		Reading					11.9	57.2	23.8
25 January 2023		Rings					10.5	28	16.6
25 January 2023		Game					9.6	28.3	26.2
25 January 2023		CRT 2					16.3	21.1	24
24 February 2023	Morning	CRT 1	385	39	395	43	205.1	886.1	33.5
24 February 2023		Reading					528.2	265.5	28.4
24 February 2023		Rings					232.9	260.1	14.5
24 February 2023		Game					303.1	338.7	22.3
24 February 2023		CRT 2					178.8	1341.5	22
24 February 2023	Afternoon	CRT 1	395	48	411	40	168.8	406.1	19.2
24 February 2023		Reading					425.5	315.9	22
24 February 2023		Rings					275.1	47.5	15
24 February 2023		Game					344.7	212.7	22.4
24 February 2023		CRT 2					233.7	1267.6	19.8
24 February 2023	Evening	CRT 1	420	57	403	31	255.6	890.8	29.2
24 February 2023		Reading					400.7	554.3	22.3
24 February 2023		Rings					213.7	481.7	18.1
24 February 2023		Game					286.3	140.7	21.8
24 February 2023		CRT 2					148.4	384.9	22.7

**Table 12 sensors-23-06197-t012:** Types of eye movement characteristics.

Characteristic Type	Number of Characteristics
Speed	12
Acceleration	6
Quantity	7
Time	7
Percentage	8
Frequency	5
Ratio	3
Length	3

**Table 13 sensors-23-06197-t013:** Correlation of characteristics with average reaction time and standard deviation of CRT task.

Characteristic	Correlation	Diameter of Eye Fixation Area
Average ratio of the path to the gaze movement	75%	1.6
Percentage of fixations is longer than 900 ms	74%	1.5
Duration of fixations is shorter than 150 ms	73%	1.2
Ultrashort fixations per minute	71%	0.5
Percentage of fixations time is shorter than 150 ms	67%	1.2
Number of fixations is shorter than 150 ms per minute	63%	0.7
Maximum eye movement speed in a second interval, degrees of visual angle per sec	62%	0.1–2.3
Number of fixations is shorter than 180 ms per minute	61%	1
Maximum frequency of appearance of a new fixation area per second, Hz	58%	1.6
Saccade number with an amplitude of less than 6 degrees of visual angle per minute	56%	2.3
Percentage of fixations is shorter than 180 ms	53%	0.5
Percentage of fixations longer than 180 ms	53%	0.5
Percentage of fixations is shorter than 150 ms	51%	0.5
Module of minimum acceleration in the second interval, degrees of visual angle per sec^2^	49%	0.1–2.3
Saccade number with an amplitude of more than 6 degrees of visual angle per minute	45%	2.2
Maximum saccade speed, degrees of visual angle per sec	39%	0.9
Maximum eye movement velocity, degrees of visual angle per sec	32%	0.1–2.3
Average saccade duration in seconds	32%	1.4
Average appearance frequency of a new fixation area per second, Hz	28%	2.3
Maximum ratio of the path to the gaze movement	27%	0.9
Minimum eye movement speed in a second interval, degrees of visual angle per sec	27%	0.1–2.3
The number of fixations longer than 180 ms per minute	23%	2.2
Maximum acceleration module in a second interval, degrees of visual angle per sec^2^	23%	0.1–2.3

**Table 14 sensors-23-06197-t014:** Results of scales and questionnaires for each of the participant.

Number	Gender	Age	PSQI	BDI-II	FAS	Field Dependence	Dominant Hand	Dominant Eye
1	m	25	5	15	18	dependent	right	right
2	f	21	7	6	21	dependent	right	right
3	m	36	11	28	39	dependent	right	right
4	m	23	6	13	21	dependent	right	left
5	f	22	6	5	18	dependent	right	right
6	f	32	5	2	20	dependent	right	right
7	f	22	7	12	25	dependent	right	right
8	f	22	6	8	24	dependent	right	left
9	m	23	6	8	21	independent	right	right
10	f	22	5	5	28	dependent	right	right
11	f	50	5	0	16	dependent	right	left
12	f	21	6	12	22	dependent	right	right

**Table 15 sensors-23-06197-t015:** Results of sleep quality assessment.

Participant	Mean Sleep Quality	Mean Time to Fall Asleep, min
1	Fairly good	35.6 (min 15, max 45)
2	Fairly good	24.2 (min 15, max 35)
3	Fairly good	21.8 (min 5, max 35)
4	Fairly bad	37 (min 30, max 60)
5	Fairly good	32 (min 10, max 80)
6	Fairly good	28.5 (min 10, max 60)
7	Fairly bad	72.2 (min 30, max 120)
8	Fairly bad	11.8 (min 5, max 25)
9	Fairly bad	20.5 (min 10, max 40)
10	Fairly bad	44.5 (min 20, max 60)
11	Fairly bad	16.5 (min 10, max 30)
12	Fairly bad	6.6 (min 3, max 10)

## Data Availability

Our dataset is available by the following link: https://disk.yandex.ru/d/KmdoodhhveGZng (accessed on 30 June 2023).

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
