# Peer review of "OperatorEYEVP: Operator Dataset for Fatigue Detection Based on Eye Movements, Heart Rate Data, and Video Information"

_sensors, 2023, doi:10.3390/s23136197_

Round 1
Reviewer 1 Report
The main question addressed by the research is to find out the signs of fatigue of workers who manage air traffic, nuclear power plant, various transport vehicles at an early stage to prevent emergency situations. The research topic is very relevant. The study contains a practical novelty.
The authors analyze the results of already conducted research in this area, however, according to the reviewer, the theoretical basis of the research is not sufficient. It would be very interesting to consider early detection systems/models developed by other authors.
The research methodology is described in detail and justified, however, the authors do not advance hypothesis(es), which would be desirable, according to the reviewer.
The results of the research require refinement, as they do not contain the results of the research on the resolution of the analyzed problem. According to the reviewer, the authors could offer their system/model early detection of fatigue in workers.
Author Response
Dear reviewer,
Thank you for valuable comments. We provide detailed answers in the attched document as well as in the papers with labelled track changes.
Authors

Reviewer 2 Report
This is an interesting article with a big amount of dat being shown. However, a small revision is suggested.
Major points:
1) Comapred to a big amount of presented data the discussion section is too short.
2) The fatigue identification dataset obtained by the Authors is rather complicated, with a big amount of data, which may be difficult for interpretation. Compared to the complexity of the recording system, importance of obtained information is rather modest . Is it reasonable to use such a complicated dataset to get informed that "lowest fatigue values consequently are in the afternoon" (page 21) or "skill to play the Tetris game improves, as does the reaction time" (page 30) ?
3) Would it be reasonable to apply such a complicated system for a routine control of fatigue of employees, such as car drivers or machine operators ? Such a control may be possible applying simple observations of human behaviour.
Minor point:
1) The Authors use a big amount of abbreviations. I think it is reasonable to make a list of used abbreviations. This would make the easier to read, especially by non-specialist readers.
Author Response

(The authors gave the same response as above.)

Reviewer 3 Report
In the current manuscript, Kashevnik et al. describe research into fatigue and how it is detected using various tools. Some data has been collected with the application of eye tracker, a video camera, a stage camera, and a heart rate monitor. I consider the research results presented in this paper to be interesting primarily from a practical point of view.
In my opinion, the manuscript was prepared correctly in terms of both content and editorial. I hereby recommend its publishing in Sensors in the current state.
Author Response

(The authors gave the same response as above.)
